# Effect of Whole-Body Cryotherapy on Iron Status and Biomarkers of Neuroplasticity in Multiple Sclerosis Women

**DOI:** 10.3390/healthcare10091681

**Published:** 2022-09-02

**Authors:** Bartłomiej Ptaszek, Szymon Podsiadło, Olga Czerwińska-Ledwig, Marcin Maciejczyk, Aneta Teległów

**Affiliations:** 1Institute of Applied Sciences, University of Physical Education in Krakow, 31-571 Krakow, Poland; 2Institute of Clinical Rehabilitation, University of Physical Education in Krakow, 31-571 Krakow, Poland; 3Institute of Basic Sciences, University of Physical Education in Krakow, 31-571 Krakow, Poland; 4Institute of Biomedical Sciences, University of Physical Education in Krakow, 31-571 Krakow, Poland

**Keywords:** whole-body cryotherapy, MS, iron, neuroplasticity, BDNF, NGF, PDGF, VEGF, IGF-1

## Abstract

The aim of the study was to compare the effect of a series of 20 whole body cryotherapy (WBC) sessions on iron levels and neuroplasticity biomarkers in women with multiple sclerosis (MS) and healthy women. Consent was obtained from the Bioethics Committee at the Regional Medical Chamber in Krakow (87/KBL/OIL/2018; 8 May 2018). The study was registered in the Australian New Zealand Clinical Trials Registry (ACTRN12620001142921; 2 November 2020). The study included 30 people: 15 women with multiple sclerosis (mean age 41.53 ± 6.98 years) and 15 healthy women (mean age 38.47 ± 6.0 years). Both groups attended cryotherapy sessions. Venous blood was collected for analysis before the WBC session and after 20 sessions. In women with MS and healthy women, no significant effect of WBC on changes in the level biomarkers of neuroplasticity was found. There were also no statistically significant differences between the groups of the analyzed indices at the beginning of the study.

## 1. Introduction

MS is a disorder of the central nervous system with different clinical and pathological features, showing different pathways of tissue damage [1]. Demyelination, inflammation and axonal degeneration are the main pathological mechanisms responsible for clinical symptoms [2]. Among central nervous system disorders, MS is the most frequent cause of permanent disability in young adults, aside from trauma [3]. The body response to whole-body cryotherapy involves changes in the hormonal, cardiovascular, nervous, muscular and immune systems [4,5]. There are studies that show an association between exposure to cold temperatures and changes in the levels of certain hormones or enzymes in patients (ACTH, beta-endorphin, cortisol, catecholamines, cytokines, uric acid, tumor necrosis factor α, adrenaline, noradrenaline, testosterone) [5,6,7,8] and there are no data on the effect of cryotherapy on neuroplasticity. In the human body, iron is stored mainly in erythrocytes (hemoglobin), liver (hemosiderin, ferritin), muscles (myoglobin), plasma (transferrin) and bone marrow [9]. According to many scientists, the impairment of iron metabolism, as well as its excessive accumulation in tissues, can provoke pathological processes [10]. The latest knowledge on the molecules related to the immune system and the CNS that shape neuroplasticity in MS relates to the brain-derived neurotrophic factor (BDNF), interleukin 1β, amyloid-β-1-42, platelet-derived growth factor (PDGF) and the cannabinoid receptor type 1 (CB1Rs) and others [11]. Neurodegenerative processes play an important role in the pathogenesis and progression of MS. Anti-inflammatory or immunomodulating therapies have limited effects on neurodegeneration and disability. That is why it is so important to look for new neuroprotective therapeutic approaches that can alleviate symptoms and prevent disease progression. The aim of this study is to compare the effect of a series of 20 whole body cryotherapy (WBC) sessions on iron levels and neuroplasticity biomarkers in women with multiple sclerosis (MS) and healthy women.

## 2. Materials and Methods

### 2.1. Participant Characteristics

The presented controlled, prospective study was carried out in accordance with the Helsinki Declaration of the World Medical Association. Ethics permission was obtained from the Bioethics Committee at the Regional Medical Chamber in Krakow (87/KBL/OIL/2018; 8 May 2018). The study was registered in the Australian New Zealand Clinical Trials Registry (ACTRN12620001142921; 2 November 2020). Participants were enrolled in the study after examination by a neurologist and a rehabilitation physician. The participants were qualified for the study after being examined by a neurologist and a rehabilitation physician.

Inclusion Criteria:Groups with MS:
○Diagnosed MS (McDonald review criteria),○Expanded Disability Status Scale (EDSS): 0 to 6.5.
All respondents (MS and CONT):
○Female sex,○Age: 30–55 years,○Written consent of the patient to participate in the study.


Exclusion Criteria—all respondents (MS and CONT):Contraindications to WBC,Change of diet during the project or immediately before,Other forms of treatments or physical activity during the project or immediately before.

The study included 30 people:Experimental group (MS): 15 women with multiple sclerosis (mean age 41.53 ± 6.98 years);Control group (CONT): 15 healthy women (without neurological diseases and other chronic diseases) (mean age 38.47 ± 6.0 years).

Both groups attended cryotherapy sessions. The characteristics of the Participants are presented in Table 1.

### 2.2. Analysis of Biochemical Blood Indices

For the analysis of blood indicators, venous blood was collected twice: before the start of WBC procedures (Study 1) and after a series of 20 WBC sessions (Study 2). Blood was collected in tubes with a clot activator in the morning from the cephalic, fallen or medial vein.

Serum iron, transferrin and ferritin concentrations were determined using a colorimetric test. The tests were performed using the Cobas c 501 analyzer.

IgG, IgA and IgM in human serum were determined by immunonephelometry using the Dade Behring device in the BN ProSpec system. A Cobas c 311/511 analyzer (Roche/Hitachi) was used for protein electrophoresis. The concentration of C-reactive protein (CRP) was assessed by immunonephelometry using reagent kits and a BN ProSpec nephelometer (Siemens Health).

BDNF, NGF, PDGF, VEGF and IGF-1 were determined in the serum. The indices were investigated with photometric tests: Human BDNF Kit, Human NGF Kit, Human PDGF Kit, Human VEGF Kit—Shanghai Sunred Biological Technology Co. and IGF-1 600 ELISA Kit—DRG Instruments GmbH, EIA 4861, Germany. The procedure was conducted in accordance with the manufacturer’s recommendations.

### 2.3. Description of the Intervention

WBC took place at the Malopolska Center of Cryotherapy and Rehabilitation in Krakow. Treatment parameters: atrium temperature: −60 °C; chamber temperature: −120 °C; refrigerant cooling: liquid nitrogen; time of a single WBC session: 1.5 min (1 treatment), 2 min (2 treatments), 3 min (3–20 treatments). Patients used the treatments once a day (15:00–17:00). Treatments were performed 5 times a week (20 in total). Each treatment was followed by a warm-up on a cyclo-ergometer for about 15 min without any resistance.

### 2.4. Statistical Analysis

Descriptive statistics were determined, mean (x) as well as standard deviation (SD). The normality of distributions was verified with the Shapiro–Wilk test. Data distribution analysis was performed using parametric tests—the Student’s *t*-test for dependent samples within the group and the same test for independent samples performing comparisons within the groups. The applied tests verified two-sided hypotheses. The analyses were performed with the use of the Statistica 13 package (Tibco Software Inc., Palo Alto, CA, USA).

## 3. Results

A statistically significant decrease was observed in Transferrin level after the whole-body cryotherapy application among MS patients (*p* = 0.042) (within normal limits—this change could be related to the exercises patients did after the WBC or to inflammation). While analyzing the remaining indicators, a favorable direction and trends of changes were observed. However, the changes were statistically insignificant. There were also no statistically significant differences between the groups of the analyzed indices at the beginning of the study (MS1/CONT1) (Table 2).

## 4. Discussion

Therapy of MS is a difficult process due to the large number of different symptoms and their overlapping. Despite a detailed knowledge about the pathomechanism of the disease, the possibilities for pharmacological treatment are quite limited. Therefore, in addition to pharmacotherapy, physiotherapy and physical activity play an important role. Neuroplasticity is the ability of the brain to adapt to environmental changes (internal and external). For researchers, the brain’s response to damage in various disease states, e.g., in patients with multiple sclerosis, is of major interest. While in healthy subjects brain plasticity is based on the development of brain structure, memory and learning processes, in the case of people with MS, it also affects changes at the molecular, synaptic and cellular levels [11]. Probing the limits of plasticity is challenging in MS because of the widespread and multifaceted nature of the disease. The aim of this study was to compare the effect of a series of 20 whole body cryotherapy (WBC) sessions on iron levels and neuroplasticity biomarkers in women with multiple sclerosis (MS) and healthy women. The incidence of MS in women compared to men in Europe is around 2:1, and the higher incidence of MS in women may be related to the higher incidence of viral diseases in this sex or more women in the population. In order to eliminate the changes related to the aging and menopause, the age of the patients was set at 30–55 years.

More and more studies in recent years have confirmed the role of iron dysregulation in the pathogenesis of MS, as iron is essential for oxidative phosphorylation and myelin formation [12]. Increased iron storage is seen in a variety of chronic neurological disorders, including MS [13]. In a study by Khalil et al. (2014), cerebrospinal fluid transferrin was decreased in MS patients compared to healthy controls, while no serum differences were observed. Altered brain iron homeostasis occurs early in the disease and suggests transferrin may play a role in brain iron deposition in people with MS [14].

There are studies showing that CSF ferritin and transferrin values can help diagnose the disease, however Gezer et al. (2021) found no significant difference in transferrin and ferritin levels in people with MS compared to healthy controls [15]. The aim of the research by Iranmanesh et al. (2013) was an evaluation of serum iron and ferritin in patients with multiple sclerosis and a healthy control group. They did not observe any difference in these groups [16]. Sfagos et al. (2005) and Abo-Krysha (2008) showed in their studies that serum iron level of patients compared to the control group had no significant difference, but serum transferrin level was higher which suggested iron dysfunction in these patients [12,17]. Some studies were published that showed abnormal serum levels of this ion. Forte in his research observed that serum iron level of the patients was higher compared to the control group (2005) [18]. Alimonti et al. (2007) and Johnson et al. (2000) obtained similar results [19,20]. In our study, we did not observe the changed mean values of transferrin, ferritin and iron in people with MS compared to healthy people; only people with MS showed a lower level of transferrin after cryotherapy. The aim of the study by Ferreira et al. (2017) was to study ferritin levels and check their relationship to the parameters of oxidative stress and MS progression. The authors concluded that ferritin may increase oxidative stress in patients with multiple sclerosis and play a role in disease progression [21].

Olsson et al. (2021) conducted a search of works in databases on the study of CRP levels in MS. The results of the studies were inconsistent, and the literature does not support the use of the CRP level as a biomarker in MS—neither diagnostic nor prognostic [22]. We did not observe any differences or changes in CRP (Table 2).

BDNF plays an important role in the processes of neuroregeneration, neuroprotection and neurogenesis, and low values may lead to accelerated disability [23]. Patanella et al. (2009) studied 30 patients with relapsing-remitting MS and found that low BDNF levels were correlated with longer time to complete the task of split attention and visual scanning [24]. Frota et al. (2009) observed decreased BDNF levels in MS patients compared to healthy controls and did not notice any correlation between mean BDNF levels and clinical parameters [25], whereas Sarchielli et al. (2002) found an inverse relationship between BDNF and EDSS levels [26]. Despite many studies on BDNF, the mechanism of the influence of low temperatures on neurotrophins is poorly studied. In a study by Zembron-Lacny et al. (2020), BDNF concentration did not change after cryotherapy treatments [27]. Other results were obtained by Rymaszewska et al. (2018), who observed an increase in BDNF and a reduction in memory deficits in people with cognitive impairment after a series of WBC treatments [28]. According to the research by Zembron-Lacny et al. (2020) levels of circulating growth factors, HGF, IGF-1, PDGF, VEGF and BDNF, were also reduced by exposure to WBC [27]. In our study, we did not observe any differences in baseline BDNF levels or changes following whole body cryotherapy. In our study, we did not observe any differences in baseline BDNF levels or changes following whole body cryotherapy (Table 2).

NGF is of fundamental importance for the differentiation, regeneration and growth of neurons in the peripheral nervous system (PNS), as well as for the proper functioning of cholinergic neurons in the CNS [29]. Moreover, it has been proved that NGF is responsible for promoting the biosynthesis of myelin sheaths [30,31]. Lower urinary tract dysfunction is a very common problem in neurological patients. More and more often, new and non-invasive diagnostic methods are sought, and neurotrophins—potential biological markers of the neurogenic bladder—are valuable in this case. The aim of the research by Bazhenov et al. (2018) was to evaluate the specificity and sensitivity of BDNF and NGF in urine and serum in MS patients as markers of detrusor overactivity. The authors concluded that NGF is a very specific biomarker, while BDNF is very sensitive. Therefore, the most valuable will be the combination of both measurements—serum NGF and BDNF [32]. In our study, we did not observe any differences in baseline NGF levels or changes following whole body cryotherapy (Table 2).

Platelet-derived growth factor (PDGF) is one of the most important factors in MS remission. It promotes neuronal differentiation, remyelination and leads to an increase in the density of oligodendrocytes [33]. The research of Mori et al. (2013) showed that PDGF was significantly lower in patients with primary progressive MS (PPMS) than in patients with RRMS, and also statistically non-significantly lower than in the control group [34]. In another article, Mori et al. (2014) showed that a high level of PDGF occurs in patients who have fully recovered, and a low level correlates with a worse clinical picture [35]. It has also been observed that the concentration of PDGF in the CSF of people with MS decreases with the duration of the disease, and its levels in the serum and CSF can be used as markers indicating the severity of the disease [36]. In our study, we did not observe any differences in baseline PDGF levels or changes following whole body cryotherapy (Table 2).

A very important aspect of the pathophysiology of MS are demyelinating changes that result from the release of angiogenic molecules, e.g., vascular endothelial growth factor (VEGF). It plays an important role in the development of neurodegenerative diseases through inflammation. Initially, VEGF acts as a pro-inflammatory primer, while later it reduces the reactivity of angiogenic molecules. It also shows increased activity in the area of neuronal damage resistance and plays a regulatory role in the proliferation, migration and differentiation of neuronal progenitor cells [37,38]. In our study, we did not observe any differences in VEGF baseline levels or changes following whole body cryotherapy (Table 2).

IGF-I likely play a fundamental role in the myelination process. Low serum IGF-1 levels correlated with cognitive impairment and fatigue in MS patients [39]. In our study, we did not observe any differences in baseline IGF-1 levels or changes following whole body cryotherapy (Table 2). Our results are similar to those of Gironi et al. (2013) and Lanzillo et al. (2011) [40,41]. In addition, Wilczak et al. (1998) found no differences in the cerebrospinal fluid and IGF-1 serum between healthy subjects and patients with multiple sclerosis [42]. However, Ghassan et al. (2017) observed a significant increase in serum IGF-1 levels in people with multiple sclerosis compared to healthy people. This difference may be due to differences in age and duration of the disease [43].

The current experiment was not without its drawbacks, related to a small number of participants, the lack of a uniform diet, or the lack of monitoring of the exact activity of the subjects and the daily schedule. The authors of the study took into account the change of diet and physical activity immediately prior to enrollment in the project and during its duration. Exercise-induced changes are measured by various methods that reveal molecular, cellular, structural, functional and behavioral changes. El-Sayes et al. (2018) summarized exercise-induced changes as a model of chronic and acute aerobic exercise-induced neuroplasticity [44]. Diet and sleep are also important [45].

## 5. Conclusions

To our knowledge, this study is the first to compare the effects of WBC on iron levels and the biomarkers of neuroplasticity in MS and healthy women. There was no significant effect of a series of systemic cryotherapy treatments on changes in iron status and neuroplasticity biomarkers in either MS or healthy women. There was also no difference between the groups. WBC seems to be a safe form of therapy. Research should continue with more patients, and randomized.

## Figures and Tables

**Table 1 healthcare-10-01681-t001:** General characteristics of the respondents.

Characteristics	MS n = 15	CONT n = 15
Age [years]	41.53 ± 6.98	38.47 ± 6.00
Body height [cm]	165.93 ± 6.53	169.4 ± 5.79
Body mass [kg]	66.75 ± 16.78	72.35 ± 13.85
Body mass index [kg/m^2^]	24.18 ± 5.68	25.22 ± 4.81
Fat [%]	33.26 ± 7.45	30.47 ± 6.65
Lean body mass [kg]	43.45 ± 5.68	49.55 ± 5.90
Total body water [kg]	31.83 ± 4.21	36.28 ± 4.32
EDSS score	3.03 ± 1.67	-
Disease duration [years]	11.00 ± 6.49	-
Disease course [%]—Primary progressive	13.33	-
Disease course [%]—Relapsing-remitting	86.67	-

**Table 2 healthcare-10-01681-t002:** The iron status, biochemical parameters and biomarkers of neuroplasticity in the experimental group (MS) and in the control group (CONT) before the intervention sessions (Study 1) and after the intervention sessions (Study 2).

Parameter	MS Study 1	MS Study 2	(*p*)	CONT Study 1	CONT Study 2	(*p*)	MS Study 1/CONT Study 1 (*p*)
Iron [µmol/L]	15.37 ± 8.76	12.99 ± 5.99	0.151	16.06 ± 6.14	13.42 ± 4.38	0.179	0.805
Ferritin [ng/mL]	37.97 ± 37.83	31.92 ± 28.95	0.244	19.22 ± 11.21	21.27 ± 14.38	0.390	0.076
Transferrin [g/L]	2.97 ± 0.44	2.78 ± 0.44	0.042	2.75 ± 0.58	2.70 ± 0.50	0.305	0.261
IgG [g/L]	11.56 ± 2.81	11.24 ± 2.37	0.092	12.73 ± 2.32	12.38 ± 2.25	0.361	0.223
IgA [g/L]	2.24 ± 0.99	2.16 ± 0.96	0.132	1.72 ± 0.40	1.76 ± 0.37	0.504	0.068
IgM [g/L]	1.33 ± 0.72	1.19 ± 0.59	0.068	1.17 ± 0.37	1.18 ± 0.33	0.806	0.456
CRP [mg/L]	2.03 ± 2.22	1.27 ± 0.79	0.19	2.60 ± 3.54	2.22 ± 2.86	0.530	0.603
bdnf [ng/mL]	1.89 ± 0.51	2.21 ± 2.13	0.562	2.40 ± 1.87	2.25 ± 1.70	0.513	0.323
ngf [pg/mL]	184.63 ± 119.09	262.60 ± 287.11	0.201	290.75 ± 280.72	262.67 ± 201.82	0.589	0.188
pdgf [pg/mL]	454.87 ± 361.69	452.97 ± 382.48	0.981	383.46 ± 231.79	450.42 ± 315.12	0.449	0.524
vegf [ng/L]	616.30 ± 304.19	649.78 ± 400.24	0.616	690.77 ± 402.40	645.46 ± 351.48	0.592	0.572
igf-1 [ng/mL]	79.50 ± 33.56	90.70 ± 37.92	0.080	90.42 ± 37.41	119.45 ± 104.12	0.327	0.407

## Data Availability

All data generated or analyzed during this study are included in this published article.

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
