# Peer review of "Effect of Whole-Body Cryotherapy on Iron Status and Biomarkers of Neuroplasticity in Multiple Sclerosis Women"

_healthcare, 2022, doi:10.3390/healthcare10091681_

Round 1

Reviewer 1 Report

Dear Authors,

First of all, I think that the manuscript entitled: “Effect of Whole-Body Cryotherapy on Iron Status and Biomarkers of Neuroplasticity in Multiple Sclerosis Women” submitted for publication in the Healthcare Journal (MDPI) has both scientific and clinical interest.

Comments:

[Introduction Section]:  

     I consider that the authors should mention (before the aim of the study) other studies in the same field (effects of whole-body cryotherapy in neuroplasticity biomarkers) if they exist (if not I consider that it will important to write not only in conclusion but and before the aim of the study in the introduction section). Also, I consider that is important for authors to explain the physiologic mechanisms that they believe/consider will affect those neuroplasticity biomarkers (as a result of whole-body cryotherapy in multiple sclerosis patients).

[Discussion Section]:

     Something that I think is particularly important is that the authors in the discussion section should refer extensively to the limitations of the study in question (mainly the limitations that arise from the biomarkers they used and more specifically whether the results of these biomarkers will potentially be influenced by other exogenous conditions or habits of the study participants (e.g. diet, sleep schedule, etc.).

Author Response

Thank you for your review and all comments.

[Introduction Section]: I consider that the authors should mention (before the aim of the study) other studies in the same field (effects of whole-body cryotherapy in neuroplasticity biomarkers) if they exist (if not I consider that it will important to write not only in conclusion but and before the aim of the study in the introduction section). Also, I consider that is important for authors to explain the physiologic mechanisms that they believe/consider will affect those neuroplasticity biomarkers (as a result of whole-body cryotherapy in multiple sclerosis patients).

- the description of the research was extended, and the justification of the research purposefulness was added

[Discussion Section]: Something that I think is particularly important is that the authors in the discussion section should refer extensively to the limitations of the study in question (mainly the limitations that arise from the biomarkers they used and more specifically whether the results of these biomarkers will potentially be influenced by other exogenous conditions or habits of the study participants (e.g. diet, sleep schedule, etc.).

- a paragraph on limitations has been added

Reviewer 2 Report

Dear authors,

First of all, the manuscript is interesting and evaluates important factors in the context of MS. However, I have some recommendations:

Introduction:
- authors state that there are studies that show an association between exposure to cold temperatures and changes in the levels of certain hormones or enzymes in patients. Any of those refers to an association between whole-body cryotherapy and iron levels / neuroplasticity biomarkers in MS or similar conditions? If yes, they should be mentioned, to help contextualize the aim of this study

- the majority of MS studies include participants of both sexes. The reason why this study only included female patients should be explained

Material and methods:
- regarding the participants, were they all under the same treatment? That should be clarified on the inclusion criteria since it is one factor that could have an impact on the results

- the naming of the columns in table 1 should be revised to be in accordance with the groups referred to in the text and the naming of the columns in table 2.

Results: require a more detailed explanation. For example, what does the decrease of transferrin level suggests or how could this result be important in the context of the patients?

Conclusion/discussion: changes in diet, as well as other forms of treatment or physical activity, are described as exclusion criteria. It would be interesting to explain how these factors could impact the study results.

Minor:

- The study included one experimental group (MS) and a control one (CONT). In table 1, instead of the CONT group, it is represented a men's group (mean age 38.47 ± 6.0 years). Since the mean age is equal to the one described for the control group, this is just one error in the group naming or do the results correspond to other data?

Author Response

Thank you for your review and all comments.

Introduction:
- authors state that there are studies that show an association between exposure to cold temperatures and changes in the levels of certain hormones or enzymes in patients. Any of those refers to an association between whole-body cryotherapy and iron levels / neuroplasticity biomarkers in MS or similar conditions? If yes, they should be mentioned, to help contextualize the aim of this study

- description added

- the majority of MS studies include participants of both sexes. The reason why this study only included female patients should be explained

 - added a statistical explanation

Material and methods:
- regarding the participants, were they all under the same treatment? That should be clarified on the inclusion criteria since it is one factor that could have an impact on the results

- description added

- the naming of the columns in table 1 should be revised to be in accordance with the groups referred to in the text and the naming of the columns in table 2.

- changed

Results: require a more detailed explanation. For example, what does the decrease of transferrin level suggests or how could this result be important in the context of the patients?

- description added

Conclusion/discussion: changes in diet, as well as other forms of treatment or physical activity, are described as exclusion criteria. It would be interesting to explain how these factors could impact the study results.

- description added

Minor:

- The study included one experimental group (MS) and a control one (CONT). In table 1, instead of the CONT group, it is represented a men's group (mean age 38.47 ± 6.0 years). Since the mean age is equal to the one described for the control group, this is just one error in the group naming or do the results correspond to other data?

- error while editing, sorry - changed

Reviewer 3 Report

First of all, I would like to thank the authors for the research effort. 

Secondly, I would like to propose some recommendations on the paper:

Introduction: it would be necessary to provide data on the importance of these aspects that are intended to be improved in patients with multiple sclerosis. The choice of the inclusion criteria for the patients listed below is not sufficiently justified. What difference does the fact that they are only women make, and why that particular age range? 

In table 1: the footnote is missing.

Table 2 requires further explanation in the text. The results are too brief.

I recommend modifying the conclusions to align them with the objectives of the study.

Author Response

Thank you for your review and all comments.

Introduction: it would be necessary to provide data on the importance of these aspects that are intended to be improved in patients with multiple sclerosis. The choice of the inclusion criteria for the patients listed below is not sufficiently justified. What difference does the fact that they are only women make, and why that particular age range? 

- new data and justification added

In table 1: the footnote is missing.

- changed

Table 2 requires further explanation in the text. The results are too brief.

- corrected

I recommend modifying the conclusions to align them with the objectives of the study.

- adjusted

Round 2

Reviewer 3 Report

Congratulations on the research. I have no further comments